# Chorioretinal Folds in the Trabeculectomized Eye with Normal Intraocular Pressure after Phacoemulsification

**DOI:** 10.3390/medicina57090896

**Published:** 2021-08-28

**Authors:** Yi-Fen Lai, Ching-Long Chen, Ke-Hao Huang, Yi-Hao Chen

**Affiliations:** 1Department of Ophthalmology, Tri-Service General Hospital, National Defense Medical Center, No. 325, Sec. 2, Chenggong Rd, Neihu Dist., Taipei City 114, Taiwan; iris92929@gmail.com (Y.-F.L.); doc30881@mail.ndmctsgh.edu.tw (C.-L.C.); a912572000@gmail.com (K.-H.H.); 2Department of Ophthalmology, Song-Shan Branch of Tri-Service General Hospital, National Defense Medical Center, No. 131, Jiankang Rd., Songshan Dist., Taipei City 105, Taiwan

**Keywords:** chorioretinal folds, trabeculectomy, cataract surgery, hypotony

## Abstract

*Background and Objectives*: This case report discusses possible causes of chorioretinal fold (CRF) formation. *Materials and Methods*: A case report. *Results*: A 48-year-old man presented with a history of high myopia and primary open-angle glaucoma in both eyes. He underwent a trabeculectomy followed by phacoemulsification in both eyes. Two months later, he complained of blurred vision in the right eye. The intraocular pressure (IOP) was 17 mmHg in the left eye and 9 mmHg in the right eye. Refraction showed a hyperopic shift in both eyes. Slit-lamp examination showed a deep anterior chamber without cells and a well-functional bleb without leakage. Fundus examination revealed CRFs in the macula of the right eye. No papilledema, choroidal lesions, or other retinal lesions were found. Wrinkling of CRFs at the macula, an increase in central foveal thickness, and a fluid cleft were demonstrated by spectral-domain optical coherence tomography. After using steroid eye drops, the IOP in the right eye and refraction in both eyes recovered to the baseline level. Visual acuity improved in both eyes. *Conclusions*: CRFs in trabeculectomized eyes with normal IOP after phacoemulsification have not been reported. This case demonstrated that the trabeculectomized eye remains at risk of CRF formation, even if the IOP is normal without hypotony. The importance of a detailed fundus examination in patients with unexplained blurred vision may be necessary after having undergone these procedures. The early recognition of the cause of visual loss may facilitate immediate treatment and may avoid irreversible changes with permanent visual loss.

## 1. Introduction

Chorioretinal folds (CRFs) are undulations of the choroid and overlying Bruch’s membrane, retinal pigment epithelium, and neurosensory retina. On fundus examination, they appear as alternating dark and light lines at the posterior pole. They are often arranged in horizontal parallel lines but are occasionally arranged in vertical, oblique, radiating, and concentric patterns. With advances in diagnostic tools, CRFs are now recognized as a clinical sign of underlying diseases rather than a specific diagnosis. Investigating the underlying causes of CRF is necessary to ensure adequate treatment.

CRFs can be attributed to ocular or extraocular causes. However, a few cases are idiopathic, i.e., no apparent cause is identified. Ocular causes include staphyloma, hypotony, intraocular tumor, hyperopia, papilledema, vascular occlusion, thyroid-related ophthalmopathy, uveitis, uveal effusion syndrome, choroiditis, posterior scleritis, choroidal neovascular membranes, and central serous chorioretinopathy [1,2,3,4]. CRFs also occur in patients with extraocular disease of intraorbital tumor, idiopathic orbital inflammation, autoimmune disease, intracranial hypertension, or Lyme disease [5,6,7]. In 1993, Leahey et al. reviewed 54 patients with CRFs and compared causes of monocular and binocular cases [8]. They reported scleritis, vascular occlusion, and intraocular tumors as common causes of monocular cases and macular degeneration, hyperopia, and idiopathic causes as common causes of bilateral cases. In the existing literature, no case of CRFs in the trabeculectomized eye with normal intraocular pressure (IOP) after phacoemulsification has been reported. This case report discusses possible causes of CRF formation.

## 2. Case Report

A 48-year-old man with a history of high myopia but no systemic disease was diagnosed with primary open-angle glaucoma in both eyes 20 years previously. Due to poor IOP control, he underwent a trabeculectomy with adjuvant mitomycin C in the left and right eyes four years previously. The IOPs of both trabeculectomized eyes were stable, ranging from 11 to 19 mmHg under a fixed combination of brimonidine 0.2% and timolol 0.5% (Combigan^®^; Allergan, Markham, ON, Canada). Subsequently, he underwent phacoemulsification in both eyes. The preoperative axial lengths (ALs) were 28.09 and 27.7 mm in the left and right eyes, respectively. The preoperative spherical equivalent (SE) refraction was −9.75 diopters (D) with best-corrected visual acuity (BCVA) of 6/20 in the left eye, and SE was −9.75 D with BCVA of 6/15 in the right eye. Treatment of monovision corrected the left eye for near vision with residual SE of −2.50 D and the right eye for distant vision without residual SE. Two months after cataract surgeries, the patient complained of blurred vision in the right eye without pain or other systemic complaint. At initial presentation, the IOP was 17 mmHg in the left eye and 9 mmHg in the right eye. The refraction showed a hyperopic shift in both eyes (SE: −1.75 and +1.50 D in left and right eyes, respectively). The extraocular movement test revealed free ocular motility. Slit-lamp examination revealed a clear cornea, normal conjunctiva, and deep anterior chamber without cells and flare, well-positioned intraocular lens, and functional bleb without leakage in both eyes. Fundus examination revealed a clear vitreous and alternating dark and light lines obliquely parallel at the macula of the right eye (Figure 1a). No papilledema, choroidal lesions, or other retinal lesions were noted. Wrinkling of CRFs at the macula, an increase in central foveal thickness compared with the baseline (287 vs. 257 nm), and a fluid cleft were demonstrated at the corresponding site by spectral-domain optical coherence tomography (OCT; Figure 1b). After treatment with topical steroid eye drops (1% econopred Plus^®^; Alcon Laboratories, Ft Worth, TX, USA) four times a day and anti-glaucoma medication cessation, the IOP in the right eye recovered to 12 to 15 mmHg 1 month later. No evidence of CRFs was seen in the fundus on OCT during the follow-up visits (Figure 2a,b). The refraction returned to the baseline level (SE: −2.50 and +0.00 D in left and right eyes, respectively) in the following 4 months. BCVA was 6/10 in both eyes.

## 3. Discussion

We reported a case of CRF formation in a trabeculectomized eye with normal IOP after phacoemulsification. The case excluded the currently known causes of CRF. The patient’s medical history and associated symptoms excluded autoimmune disease, intraorbital tumor, idiopathic orbital inflammation, thyroid-related ophthalmopathy, posterior scleritis, Lyme disease, and intracranial hypertension. An auto-refractometer excluded hyperopia. Fundoscopic examination excluded papilledema, choroidal mass, and choroiditis. OCT excluded choroidal neovascular membranes and central serous chorioretinopathy. To the best of our knowledge, no case of CRFs in trabeculectomized eyes with normal IOP after phacoemulsification has been reported.

The clear cut of CRFs formation occurred after phacoemulsification possibly because of subclinical inflammation due to phacoemulsification. The blood–ocular barrier separates the inner portion of the eye from blood that enters the eye. Turbulence and thermal injury in phacoemulsification break down the blood–ocular barrier and increase the retinal permeability. Aqueous flare and macular edema are associated with postoperative inflammation. In a previous study, subclinical macular edema was reported after uncomplicated cataract surgery [9]. Changes in macular thickness could last 6 months postoperatively [10,11]. These subclinical changes are inflammatory responses of the immune system [12]. Inflammation promotes extravasation of fluid within the choroidal and retinal tissues. The fluid accumulated at the focal retina and choroidal tissue, while the surrounding tissue remained unchanged. An uneven tension force led to CRFs [13,14]. In the present case, although there was no aqueous cell or flare seen, an increase in the central foveal thickness and fluid cleft were revealed by OCT. After steroid use, CRFs regressed. Although we did not perform fluorescent angiography (FAG) or indocyanine green angiography (ICG), the evidence of OCT could not exclude post-phacoemulsification inflammation-induced CRFs.

The second possible cause is a decrease in scleral rigidity due to high myopia. In the present case, CRFs occurred at a relatively lower IOP compared to baseline. The relationship between IOP and biomechanical properties of the ocular structure could be a determining factor of CRFs. Myopic eyes are associated with reduced extracellular matrix, particularly at the posterior pole of the eye [15]. Loss of scleral tissue during the development of axial myopia contributes to the weakened biomechanical properties of the sclera [16]. Sergienko et al. measured the AL of the eye before and during the application of external pressure of 30 g on the eye [17]. They found that the hypermetropic and emmetropic eyes possessed a stiff sclera [17]. The extent of AL remained practically unchanged during IOP elevation in these eyes. A higher degree of myopia is associated with increased AL elongation under external pressure [17]. This proved that more myopic eyes possessed a weaker sclera [17,18]. Any cause of IOP lowering might add insult to injury in a weakened sclera. The sclera in patients with high myopia may not be sufficiently strong to maintain the contour of the posterior globe and make the globe similar to a deflated balloon [19]. Thus, tractional forces exerted by the inferior oblique muscle at its insertion point located behind the macular area can result in CRFs [20]. In addition, CRFs occurred at a relatively low IOP. Hyperopic shift was noted after cataract surgery, which indicated a decrease in the anterior–posterior diameter. CRFs disappeared after IOP returned to the baseline level. Clinical findings were similar to those of hypotony maculopathy. Interestingly, IOP in our case was within the normal range. To explain this, we postulated two possible mechanisms. First, the IOP sufficient to maintain the corneal structure might not be sufficiently strong to maintain the posterior segment structure in a patient with high myopia because high myopia is characterized by scleral thinning and localized ectasia of the posterior sclera [21]. Therefore, we could obtain a normal IOP; however, the posterior segment is presented similarly to hypotony maculopathy. Second, the IOP at presentation might be higher than the actual average IOP. The average IOP in days prior to the clinic visit and the diurnal fluctuation of IOP might have been much lower than the IOP at presentation.

Our study had several limitations. First, we did not perform FAG or ICG to observe the entire appearance of the retina. Tiny choroidal and retinal lesions could have been missed on OCT. In addition, without FAG or ICG, we could only infer subclinical inflammation based on the increased macular thickness and fluid cleft on OCT. Second, we did not prove that the structure of the posterior segment is more vulnerable to collapse than that of the cornea with a relatively low IOP. A few studies have compared the biomechanical properties of different eye structures. Third, we did not have the IOP information prior to the presentation or measure the range of diurnal fluctuations in IOP. Fourth, this was merely a case report. Further studies are required to confirm our conclusions.

## 4. Conclusions

We described a case of CRF formation in trabeculectomized eyes with normal IOP after phacoemulsification, which has not been reported before. Our case suggests that the trabeculectomized eye remains at risk of CRF formation, even if the IOP is normal without hypotony. Fundus examination is recommended for patients with blurred vision. Early recognition of the cause and immediate treatment may avoid irreversible changes with permanent visual loss.

## Figures and Tables

**Figure 1 medicina-57-00896-f001:**
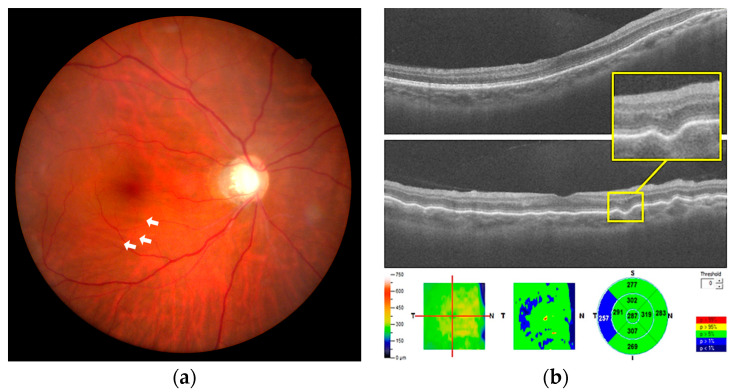
(**a**) Fundus color photography shows chorioretinal folds, appearing as faint alternating dark and light lines at the macula of the right eye (white arrows). (**b**) Wrinkling of chorioretinal folds at the macula and a small fluid cleft between the outer nuclear layer and the outer plexiform layer (yellow square) are noted at the same corresponding site on spectral-domain optical coherence tomography.

**Figure 2 medicina-57-00896-f002:**
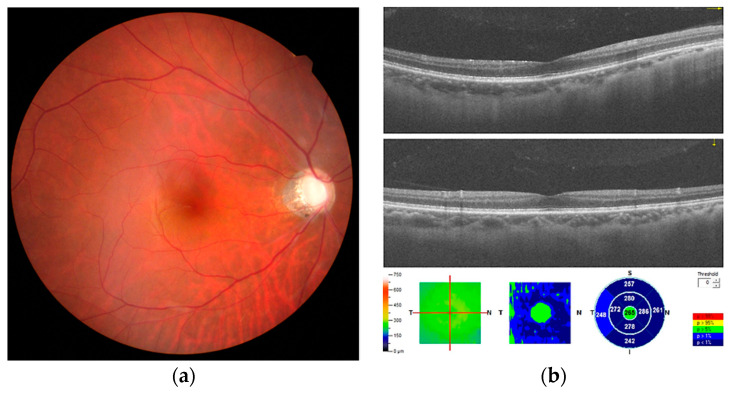
(**a**) The chorioretinal folds have regressed 1 month later. (**b**) The chorioretinal folds have regressed after intraocular pressure returns to the baseline level with the steroid treatment.

## Data Availability

The data supporting the conclusions of this article are included within the article.

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
