# Peer review of "Chorioretinal Folds in the Trabeculectomized Eye with Normal Intraocular Pressure after Phacoemulsification"

_medicina, 2021, doi:10.3390/medicina57090896_

Round 1

Reviewer 1 Report

I suggest a thorough proof reading by an English native speaker.

Authors hypothesize a postsurgical (after cataract phacoemulsification) inflammatory cause for CRF onset in the patient RE. It seems quite plausible. Moreover, CRF resolution after topical steroid treatment supports even more an inflammatory cause.

At presentation IOP in RE was 9 mmHg – it cannot be excluded that in previous days IOP could have been “very low”

Line 58: … 0.5% fixed combination (Combigan ….

Reviewer 2 Report

Thank you for the opportunity to review the manuscript. This case is interesting, but several minor changes are needed.

  1. I could not see CRFs on a color photograph in Figure 1. Please revise Figure 1 (A)so that CRFs are clearly seen.
  2. Diurnal fluctuation of intraocular pressure might be associated with the development of CRFs. Did the authors measure diurnal fluctuation of intraocular pressure in this case?
  3. Was the topical steroid eye drops effective for lowing the IOP in this case?
